# Considering the Possible Role of Pharmacists According to the Presence or Absence of Lifestyle-Related Diseases at the Time of Coronary CT Examination and Trends of Medication Use for These Diseases by Medical Doctors

**DOI:** 10.3390/pharmacy12040099

**Published:** 2024-06-27

**Authors:** Erika Miura-Takahashi, Kohei Tashiro, Yuhei Shiga, Yuto Kawahira, Sara Higashi, Yuki Otsu, Hidetoshi Kamimura, Shin-ichiro Miura

**Affiliations:** 1Department of Pharmacy, Fukuoka University Hospital, Fukuoka 814-0180, Japan; e.miura.ux@adm.fukuoka-u.ac.jp (E.M.-T.); yukifukuyaku@fukuoka-u.ac.jp (Y.O.); kamisan@fukuoka-u.ac.jp (H.K.); 2Department of Cardiology, Fukuoka University School of Medicine, Fukuoka 814-0180, Japan; kohei.t1027@gmail.com (K.T.); yuheis@fukuoka-u.ac.jp (Y.S.); kawahira0812@gmail.com (Y.K.); sara3034@icloud.com (S.H.)

**Keywords:** coronary artery computed tomography angiography, lifestyle-related disease, coronary artery disease, pharmacists

## Abstract

Background: Because patients often already have coronary artery disease (CAD) at the time of a coronary artery computed tomography angiography (CCTA) examination, we examined the medications prescribed by medical doctors for lifestyle-related diseases and investigated what possible role pharmacists can play in prescribing. Methods: Patients (n = 1357) who underwent CCTA examination were divided into two groups according to the presence or absence of lifestyle-related diseases [hypertension (HTN), dyslipidemia (DL) and diabetes mellitus (DM)], and the relationship between the presence or absence of CAD was examined. Results: The rate of CAD was significantly higher in patients with HTN, DL or DM than in patients without these diseases. The blood pressure in the HTN group was 140 ± 20/79 ± 13 mmHg, the low-density lipoprotein cholesterol value in the DL group was 119 ± 35 mg/dL, and the hemoglobin A1c value in the DM group was 7.0 ± 1.1%, all of which were poorly controlled. Anti-hypertensive drugs were used at low rates in the HTN group, statins were used in 47% and ezetimibe was used in 4% of the DL group, and dipeptidyl peptidase-4 inhibitors were used in 41% of the DM group. Conclusions: Since the rate of CAD (+) was high and control of the three major lifestyle-related diseases was poor, pharmacists should advise medical doctors to use combinations of drugs.

## 1. Introduction

Cerebro-cardiovascular diseases are the second-most important cause of death in Japan after cancer. Among these diseases, the prevention of myocardial infarction is particularly important, and medical professionals must strive to detect and treat coronary artery disease (CAD) early as well as to prevent its onset. Many patients are diagnosed with CAD at the time of screening for CAD using coronary artery computed tomography angiography (CCTA). CCTA is widely available in most general hospitals, particularly in Japan, and has been shown to be a useful non-invasive method worldwide. The prevalence of CAD in all patients who have undergone CCTA is 50% [1,2]. Recently, multidisciplinary team medical care involving not only medical doctors but also nurses, pharmacists, physical therapists, and other professionals has been shown to play an important role in patient treatment. It is important that people do not suffer from lifestyle-related diseases, but once a lifestyle-related disease develops, other medical professionals must assist medical doctors to effectively control it. Pharmacists must make suggestions so that medical doctors can prescribe the appropriate medication(s) for controlling lifestyle-related diseases [hypertension (HTN), dyslipidemia (DL) and diabetes mellitus (DM)] by using the most effective kinds and/or doses to suppress the onset of CAD.

Patients often have lifestyle-related diseases (HTN, DL and DM) at the time of CCTA [1,2,3,4]. It is the role of medical doctors to prescribe medicine in a medical setting. As times are changing, the pharmacist’s role has to also change. The role of pharmacists is evolving and expected to expand. Pharmacists may play a meaningful role in the primary and secondary prevention of cardiovascular diseases. In addition to dispensing medication, pharmacists can suggest more direct interventions to support the physician’s work, to improve medication adherence, to achieve the goals of desired therapeutic outcomes and to improve safe medication use and humanistic control [5,6]. A meta-analysis of 39 randomized controlled trials comprising 14,224 patients demonstrated improvements in blood pressure (BP) following pharmacists’ interventions [7]. Pharmacists often make suggestions so that medical doctors can prescribe the medication that is expected to be most effective for patients, in the most appropriate kinds and doses. On the other hand, it is not easy for pharmacists to suggest prescriptions for the appropriate control of lifestyle-related diseases. Up until now, the main role of pharmacists was to prepare medicines according to medical doctors’ instructions and to explain to patients how to take them properly and their side effects. However, the scope of a pharmacist’s work has expanded to include providing prescription advice to medical doctors. The difference between this and the general prescription proposal work by pharmacists is that information is disseminated by the pharmacist, but also evidence-based information is presented, and pharmacists’ recommendations should be clearly made based on the available information. In addition to sharing evidence with medical doctors, pharmacists should provide specifics on the context behind the presentation of that evidence, what pharmacists think based on the evidence, and what treatments are currently recommended. Therefore, in the future, pharmacists will need to accumulate and consider not only pharmacological information but also medical information.

We have been searching for new coronary risk factors using data from patients who underwent CCTA at Fukuoka University Hospital (FU-CCTA registry) [1,2,8,9,10,11]. For example, the liver fibrosis score may be an independent predictor of major adverse cardiovascular events (MACEs: cardiovascular death, ischemic stroke, acute myocardial infarction and coronary revascularization) in hypertensive patients undergoing CCTA [8]. The plasma level of pentraxin-3 was shown to be a predictor of MACEs in males, whereas smoking, but not pentraxin-3, was a predictor of MACEs in females [9]. In addition, the psoas major muscle index may be an imaging marker for evaluating the presence and/or severity of CAD in males, and particularly in non-elderly populations [10]. Finally, the total cholesterol efflux capacity in addition to high-density lipoprotein cholesterol (HDL-C), but not % cholesterol efflux capacity, was associated with the presence of MACEs [11]. In the present study, we sought to clarify what kind of prescribing guidance pharmacists should give to medical doctors to prevent CAD before CCAT examination for patients undergoing CCTA to screen for CAD. We examined the medications prescribed by medical doctors for lifestyle-related diseases and investigated what possible role pharmacists can play in prescribing.

## 2. Materials and Methods

### 2.1. Subjects

In this cross-sectional study, the subjects consisted of 1357 patients who underwent CCTA examination at our university hospital. The patients were divided into two groups based on the presence or absence of CAD, and the presence or absence of various lifestyle-related diseases (HTN, DL or DM). The relationships between the presence or absence of CAD and these three lifestyle-related diseases were examined. This study was approved by the ethics committee of Fukuoka University Hospital. The study was explained to the patient using a consent form, written consent was obtained, and then data were collected and analyzed.

### 2.2. Measurement of Coronary Stenosis by CCTA

We measured coronary stenosis by CCTA [A 64-MDCT on an Aquilion 64 (TOSHIBA, Tokyo, Japan) or 320-MDCT on an Aquilion ONE ViSION (TOSHIBA)] as previously described [1,12]. The region of interest was placed within the ascending aorta, and the scan was started when the CT density reached 100 Hounsfield Units higher than the baseline CT density. The scan was performed between the tracheal bifurcation and diaphragm. In all patients, 15 coronary artery segments were assessed using CCTA. We defined narrowing of the normal contrast-enhanced lumen by ≥50% in multiplanar reconstructions and cross-sectional images as significant stenosis. Patients with significantly stenosed coronary vessels were diagnosed with CAD.

### 2.3. Assessments of Patient Characteristics including Coronary Risk Factors, Medications, Etc.

Age, gender, body mass index (BMI), coronary risk factors [family history of cardiovascular diseases, smoking history (current and past), presence of HTN, DL and DM, estimated glomerular filtration rate (eGFR), and metabolic syndrome (MetS)], medication history [angiotensin converting enzyme inhibitor/angiotensin II receptor blocker (ACEI/ARB), calcium channel blocker (CCB), β-blocker, diuretic, statin, eicosapentaenoic acid (EPA), ezetimibe, fibrate, sulfonylurea (SU), α-glucosidase inhibitor (α-GI), biguanide, thiazolidine, dipeptidyl peptidase-4 inhibitor (DPP-4I), glucagon-like peptide-1 (GLP-1) and insulin], systolic BP (SBP), diastolic BP (DBP), serum lipid profile [low-density lipoprotein cholesterol (LDL-C), HDL-C and triglyceride (TG)], hemoglobin A1c (HbA1c), fasting blood sugar (FBS), and brachial-ankle pulse wave velocity value (baPWV, left and right average value) were collected from the patient database. BMI was calculated as weight (kg)/height (m)^2^. SBP and DBP were measured as the mean of two readings obtained in an office setting by the conventional cuff method using a mercury sphygmomanometer after 5 min of rest.

Patients who were receiving antihypertensive medication or who had SBP/DBP ≥ 140/90 mmHg were considered to have HTN [13]. Patients who were receiving lipid-lowering therapy or with TG ≥ 150 mg/dL, LDL-C ≥ 140 mg/dL, and/or HDL-C < 40 mg/dL were considered to have DL [14]. DM was defined by the American Diabetes Association criteria [15] or treatment with a glucose-lowering drug. Chronic kidney disease (CKD) was defined as proteinuria and/or an eGFR of <60 mL/min/1.73m^2^. MetS was diagnosed according to the criteria published by the Examination Committee of Criteria for Diagnosis of Metabolic Syndrome in Japan in 2005 as follows: visceral fat area ≥ 100 cm^2^, and the presence of more than two of the following: HTN (SBP 130 mmHg or DBP 85 mmHg or receiving an anti-hypertensive drug), DL (TG ≥ 150 mg/dL or HDL-C < 40 mg/dL or receiving a lipid-lowering drug) and DM [high fasting glucose (fasting glucose 110 mg/dL or receiving a glucose-lowering drug).

### 2.4. Measurement of baPWV

baPWV was measured using a volume-plethysmographic device and ankle-brachial pressure index (Colin Co., Aichi, Japan). The subject was examined in the supine position, with electrocardiogram electrodes placed on both wrists, a microphone for detecting heart sounds placed on the left edge of the sternum, and cuffs wrapped on both the brachia and ankles. The cuffs were connected to a plethysmographic sensor that determined the volume pulse form and an oscillometric pressure sensor that measured BP. Volume waveforms for the brachium and ankle were stored, and the sampling time was 10 s with automatic gain analysis and quality adjustment. The time interval between the wave front of the brachial waveform and that of the ankle waveform was defined as the time interval between the brachium and ankle (∆Tba). The distance between sampling points of baPWV was calculated automatically according to the height of the subject. The path length from the suprasternal notch to the brachium (Lb) and from the suprasternal notch to the ankle (La) was automatically obtained based on the subject’s height. The following equation was then used to obtain baPWV: baPWV = (La − Lb)/∆Tba. In all studies, baPWV was obtained after at least 5 min of rest.

### 2.5. Statistical Analysis

All of the data analyses were performed using the Excel 2019 (SSRI, Tokyo, Japan) and the SAS (Statistical Analysis System) Software Package (Ver. 9.4, SAS Institute Inc., Cary, NC, USA) at Fukuoka University. Descriptive data are presented as the mean ± standard deviation. Categorical and continuous variables were compared between groups using the *t*-test and chi-square test, respectively. *p* < 0.05 was considered significant.

## 3. Results

### 3.1. Patient Characteristics in All Patients and in the Absence or Presence of CAD

Table 1 shows the patient characteristics in all patients (n = 1357), and in the absence (n = 678) or presence (n = 668) of CAD. The average age of all patients was 66 ± 12 years, the percentage of males was 49%, and 49% of the patients had CAD (+) (Table 1). The age in the CAD (-) group was 62 ± 13 years, the percentage of males was 41%, and 30% of the participants smoked, whereas the age in the CAD (+) group was 69 ± 10 years, the percentage of males was 57%, and 39% of the participants smoked. The CAD (+) group was significantly older and had higher % male, % smoking, % CKD, and % MetS values in addition to % HTN, % DL and % DM than the CAD (-) group. On the other hand, there were no significant differences in % family history, BMI or mean baPWV between the two groups.

### 3.2. Medications of All Patients and in the Absence or Presence of CAD

As shown in Table 2, the % ACEI/ARB and % CCB values in all patients were 36% and 39%, respectively. The CAD (+) group had significantly higher % ACEI/ARB, % CCB, % β-blocker, % statin, % SU, % α-GI, % biguanide and % DPP-4I values than the CAD (-) group (Table 2). There were no significant differences in % diuretic, % EPA, % ezetimibe, % fibrate, % thiazolidine, % GLP-1 or % insulin between the CAD (-) and CAD (+) groups. There were significant differences in anti-hypertensive medicine (% ACEI/ARB, % CCB and % β-blocker) between the two groups.

### 3.3. The % CAD in the Absence or Presence of HTN, DL and DM

The percentages of patients with and without HTN were 65% and 35%, respectively. The percentages of patients with and without DM or DL were 25% and 75% or 68% and 32%, respectively. The % CAD in the absence or presence of HTN, DL and DM is shown in Figure 1. The % CAD in the presence of HTN (58%), DL (54%) and DM (63%) was significantly higher than that in the absence of these (35%, 41% and 45%), respectively (Figure 1).

### 3.4. Control Conditions of HTN, DM and DL

The average SBP and DBP in the HTN group were 140 ± 20 and 79 ± 13 mmHg, respectively; these values were significantly higher than those in the non-HTN group (129 ± 19/77 ± 12 mmHg), even though patients in the HTN group were treated with antihypertensive drugs (Figure 2). The average HbA1c and FBS values in the DM group were also significantly higher than those in the non-DM group (HbA1c, 7.0 ± 1.1% vs. 5.7 ± 0.6%; FBS, 100 ± 15 mg/dL vs. 132 ± 43 mg/dL). The average values of LDL-C and TG in the DL group were 119 ± 35 mg/dL and 154 ± 106 mg/dL, respectively. The LDL-C and TG values in the DL group were significantly higher, and the HDL-C value was significantly lower than those in patients without DL (LDL-C, 119 ± 35 mg/dL vs. 106 ± 22 mg/dL; TG, 154 ± 106 mg/dL vs. 93 ± 38 mg/dL; HDL-C, 55 ± 16 mg/dL vs. 62 ± 16 mg/dL). Thus, in the HTN, DM and DL groups, BP, HbA1c, FBS, LDL-C and TG were not well controlled.

### 3.5. Medications for HTN, DM and DL

In Table 3, medications included ARB (52%), CCB (58%), a two-drug combination of ARB and CCB (34%), β-blocker (13%) and diuretic (13%) in the HTN group, statin (47%) and ezetimibe (4%), a two-drug combination of statin and ezetimibe (only 2%), EPA (4%) and fibrate (2%) in the DL group, and SU (25%), α-GI (9%), biguanides (25%), thiazolidine (41%) in the DM group (Table 3). Although DPP-4 inhibitors were used in 41% of the patients, only 14% received a two-drug combination of biguanide and DPP-4I.

## 4. Discussion

In this study, we first confirmed that about 50% of cases are diagnosed with CAD at the time of screening for CAD using CCTA. Second, the three major lifestyle-related diseases were not well controlled in the patients studied, even though the patients were taking medications. Since the percentage of patients using a combination of two drugs was low for each disease, pharmacists should advise medical doctors to use combinations of drugs for the treatment of lifestyle-related diseases. Since this report examined the possible roles of pharmacists against medical doctor’s prescriptions, pharmacists should study not only pharmacological information but also medical information.

According to the Japanese Society of Hypertension Guidelines 2019 [13], the target BP values in the office are less than 130/80 mmHg in adults under 75 years of age, and less than 130/80 mmHg even in patients with CAD if there is no myocardial ischemia. In this study, the average SBP/DBP in the HTN group was 140/79 mmHg, whereas 52% of patients in the HTN group were taking ACEI/ARB and 58% were taking CCB (two-drug combination 34%). Few patients received combinations of these drugs for HTN. Recently, the problem of clinical inertia has attracted attention in patient treatment [16]. Clinical inertia refers to the situation in which patients continue to use antihypertensive medications as they are, without intensifying treatment due to familiarity, even though BP reduction is insufficient. The present results suggest that clinical inertia may be a cause of poor BP control. In the doctor–patient relationship, it is necessary to mutually recognize the need for further treatments based on a relationship of trust. Contributing factors include issues regarding the patient and/or physician, the medical system, and socio-economic issues. Therefore, pharmacists should keep clinical inertia in mind while understanding BP values and patient conditions when giving medication guidance to patients, explain the need for strict BP control, and provide guidance on lifestyle modifications. In addition, by using a model for predicting the onset of arteriosclerotic diseases using the Hisayama-cho risk score, we should evaluate not only BP values but also other risk factors to reduce the risk of developing arteriosclerotic cardiovascular diseases in the future [14], and pharmacists can encourage patients to take intensive antihypertensive treatment. Additionally, physicians should proactively advise patients on how to optimize the administration of antihypertensive medications. In such cases, pharmacists may be able to give appropriate advice to medical doctors, so they should be proactive in making suggestions.

The results of treatment for lifestyle-related diseases indicate that both DL and DM have the same results as HTN regarding the control of each disease. Regarding DL, even though patients with DL received lipid-lowering drug therapy, the LDL-C and TG values were significantly higher than those in patients without DL. The control target level of LDL-C for primary prevention is less than 160 mg/dL, less than 140 mg/dL, and less than 120 mg/dL for patients at low, intermediate, and high coronary risks, respectively [14]. The patients in this study were at high risk because they were suspected of having CAD and underwent CCTA, and the goal is to have an LDL-C of less than 120 mg/dL. However, the average value of LDL-C was 119 mg/dL, which indicates that it was poorly controlled. Statins are the standard medication for DL [14]. Statins suppress cholesterol synthesis and are therefore known to increase cholesterol absorption [17,18]. Therefore, it is said that the combination of statins with ezetimibe, which is a cholesterol absorption inhibitor that acts in the small intestine, is very effective [19,20]. However, the combined use of these two drugs was noted in only 2% of cases. This study included many patients at a time when a single-pill combination containing statins and ezetimibe was not available in Japan. Recently, a single-pill combination of statins and ezetimibe has been released and is widely used. Pharmacists should encourage the use of a single-pill combination therapy, although the present results cannot necessarily be compared with results after a single-pill is launched.

The average HbA1c value in the DM group was 7.0%. DPP-4I was used in 41% of the patients, and a two-drug combination consisting of DPP-4I and biguanide was used in only 14%. DPP-4I in combination with metformin, which is an oral antidiabetic drug that is classified as a biguanide, is an efficient, safe and tolerable combination therapy for type 2 diabetes [21]. This study included many patients at a time when a single-pill combination containing DPP-4I and biguanides was not available. Pharmacists should actively recommend this single-pill combination to medical doctors because it is thought to have benefits such as the effects of both ingredients and improved adherence through the combination of two ingredients into one formulation. Furthermore, sodium-glucose cotransporter-2 inhibitors (SGLT-2I) have frequently been used, and it has been reported that the combination of SGLT-2I with DPP-4I and/or biguanide is also useful [22]. Pharmacists should take these findings into consideration when giving advice. On the other hand, in Japan, single-pill combination tablets that combine two drugs (such as statins and ezetimibe, or DPP-41 and biguanide, etc.) were not available on the market at that time. While it was possible to administer two drugs in combination, if a single-pill combination tablet had been available, administration methods that combined two drugs may have become more popular.

In this study, there was no significant difference in baPWV between the CAD (-) and CAD (+) groups. PWV is a non-invasive method that is useful for screening the severity of arteriosclerosis [23]. There is no clear reason why there was no significant difference in baPWV between the two groups in this study. One possible reason is that the subjects in this study were subjects for primary screening for CAD, and included patients in whom arteriosclerosis had not yet progressed to that extent. In addition, studies limited to hypertensive patients have reported that it can predict cardiovascular risk [23,24]. Therefore, in this study, similar results may have been obtained if only hypertensive patients were analyzed. In addition, longitudinal studies have directly demonstrated that arterial stiffness as measured through PWV is an independent predictor of cardiovascular mortality [25,26]. This study was a cross-sectional study, as it did not consider the prognosis of patients. The usefulness of PWV may become clearer if the prognosis of patients is considered.

This study has several limitations. This was a single-center study. It did not consider the duration of illness with various lifestyle-related diseases or the duration of administration or dosage of various medications. It included patients at a time when a single-pill combination tablet was not yet available in Japan. In addition, this study did not involve pharmacists intervening in any way with medical doctors’ prescriptions. Since this is a cross-sectional study, this report examined the possible roles of pharmacists with regard to their prescriptions. A large-scale prospective study will be needed to address these issues.

## 5. Conclusions

The % CAD at the time of CCTA examination was high (49%), and the three major lifestyle-related diseases were not well controlled. Furthermore, the percentages of patients who used a two-drug combination in the HTN (+), DL (+) and DM (+) groups were 34%, 2% and 14%, respectively. Thus, the percentage of patients who used a two-drug combination was low, implying that pharmacists should proactively suggest the use of combination tablets to medical doctors. This study also suggests the need for improvements to the entire system by medical staff, including patient guidance and advice to medical doctors by pharmacists.

## Figures and Tables

**Figure 1 pharmacy-12-00099-f001:**
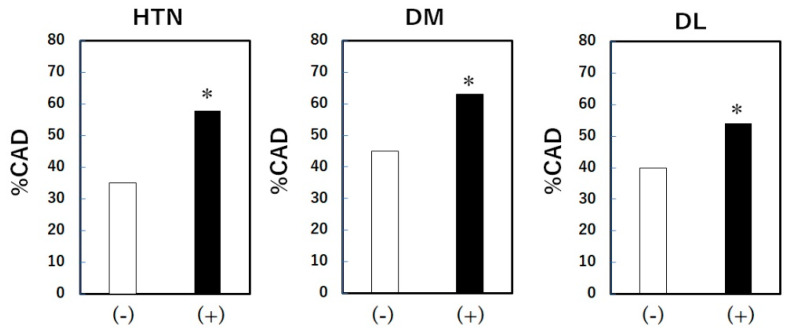
The % CAD in the absence or presence of HTN, DM and DL. CAD, coronary artery disease; HTN, hypertension; DL, dyslipidemia; DM, diabetes mellitus. * *p* < 0.05 vs. (-).

**Figure 2 pharmacy-12-00099-f002:**
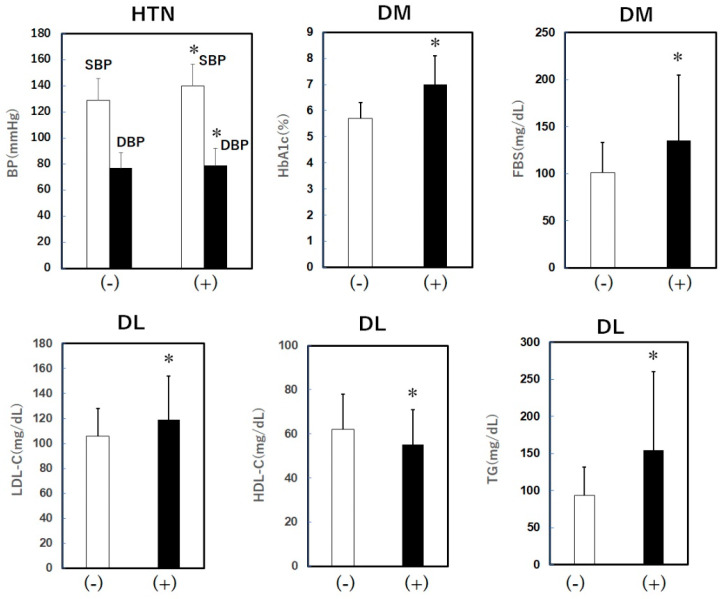
SBP and DBP in the absence or presence of HTN, HbA1c and FBS in the absence or presence of DM, and LDL-C, HDL-C and TG in the absence or presence of DL. HTN, hypertension; SBP and DBP, systolic and diastolic blood pressure; DM, diabetes mellitus; HbA1c, hemoglobin A1c; FBS, fasting blood sugar; DL, dyslipidemia; TG, triglyceride; HDL-C, high-density lipoprotein cholesterol; LDL-C, low-density lipoprotein cholesterol. * *p* < 0.05 vs. (-).

**Table 1 pharmacy-12-00099-t001:** Patient characteristics in all patients, and in the CAD (-) and CAD (+) groups.

		All Patients	CAD (-) Group	CAD (+) Group	CAD (-) Vs. CAD (+)
		(n = 1357)	(n = 678)	(n = 668)	*p*-Value
Age	yrs.	66 ± 12	62 ± 13	69 ± 10	<0.01
Gender (male)	%	49	41	57	<0.01
Family history	%	22	23	22	0.466
Smoking	%	34	30	39	<0.01
BMI	kg/m^2^	24 ± 4	24 ± 4	24 ± 4	0.348
HTN	%	65	55	76	<0.01
DM	%	25	19	32	<0.01
DL	%	68	62	74	<0.01
CKD	%	27	23	32	<0.01
eGFR	mL/min/1.73 m^2^	68 ± 16	70 ± 15	66 ± 17	<0.01
MetS	%	25	19	31	<0.01
Mean baPWV	cm/sec	1635 ± 355	1626 ± 368	1647 ± 342	0.292

CAD, coronary artery disease; BMI, body mass index; HTN, hypertension; DM, diabetes mellitus; DL, dyslipidemia; CKD, chronic kidney disease; eGFR, estimated glomerular filtration rate; MetS, metabolic syndrome; baPWV, brachial-ankle pulse wave velocity.

**Table 2 pharmacy-12-00099-t002:** Medications of all patients, and in the CAD (-) and CAD (+) groups.

Medications		All Patients	CAD (-) Group	CAD (+) Group	CAD (-) Vs. CAD (+)
		(n = 1357)	(n = 678)	(n = 668)	*p*-Value
ACEI/ARB	%	36	29	43	<0.01
CCB	%	39	31	47	<0.01
β-blocker	%	9	6	11	<0.01
Diuretic	%	8	7	10	0.091
Statin	%	32	24	39	<0.01
EPA	%	3	2	3	0.409
Ezetimibe	%	2	3	2	0.482
Fibrate	%	1	1	1	0.449
SU	%	6	4	9	<0.01
α-GI	%	2	1	3	0.027
Biguanide	%	6	5	8	0.017
Thiazolidine	%	2	2	3	0.479
DPP-4I	%	10	8	13	0.002
GLP-1	%	0.3	0.3	0.3	1.000
Insulin	%	3	2	3	0.731

CAD, coronary artery disease; ACEI/ARB, angiotensin-converting-enzyme inhibitor/angiotensin II receptor blocker; CCB, calcium channel blocker; EPA, eicosapentaenoic acid; SU, sulfonylurea; α-GI, α-glucosidase inhibitor; DPP-4I, dipeptidyl peptidase-4 inhibitor; GLP-1, glucagon-like peptide-1.

**Table 3 pharmacy-12-00099-t003:** Medications in the HTN (+), DL (+) and DM (+) groups.

Medications		HTN (+)	Medications		DL (+)	Medications		DM (+)
		(n = 883)			(n = 913)			(n = 342)
ACEI/ARB	%	52	Statin	%	47	SU	%	25
CCB	%	58	EPA	%	4	α-GI	%	9
β-blocker	%	13	Ezetimibe	%	4	Biguanide	%	25
Diuretic	%	13	Fibrate	%	2	Thiazolidine	%	41
ARB + CCB	%	34	Statin + Ezetimibe	%	2	DPP-4I	%	41
						GLP-1	%	1
						Insulin	%	10
						Biguanide + DPP-4I	%	14

HTN, hypertension; ACEI/ARB, angiotensin-converting-enzyme inhibitor/angiotensin II receptor blocker; CCB, calcium channel blocker; DL, dyslipidemia; EPA, eicosapentaenoic acid; DM, diabetes mellitus; SU, sulfonylurea; DPP-4I, dipeptidyl peptidase-4 inhibitor; GLP-1, glucagon-like peptide-1.

## Data Availability

The data that support the findings of this study are available from the corresponding author upon reasonable request.

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
