# Peer review of "Considering the Possible Role of Pharmacists According to the Presence or Absence of Lifestyle-Related Diseases at the Time of Coronary CT Examination and Trends of Medication Use for These Diseases by Medical Doctors"

_pharmacy, 2024, doi:10.3390/pharmacy12040099_

Round 1
Reviewer 1 Report
Comments and Suggestions for Authors
This is interesting brief report. Manuscript is well written. Title suggests that the role of pharmacists will be directly analyzed in presented study. Similarly, the aim of the study covers such suggestion. But the methods and results are not concentrated on that problem. This is study on risk factors control, so the results may give only a suggestion what may be improve in primary/secondary prevention.
Is it retrospective registry study? This should be clarified in methods section.
It will be interesting to see follow up data and how the CT result influence the further pharmacotherapy.
Tables and figures are informative and well prepared.
Conclusions: no sure if phrase: “suggesting the need for pharmacists to proactively suggest the use of combination tablets to medical doctors” is optimal. This suggest need for whole system improvement including the pharmacists.
The references are appropriate.
Author Response
Response to Reviewer 1
Ref. No. pharmacy-2969262R1
Presence of lifestyle-related diseases at the time of coronary artery CT examination and trends in medication use: Role of pharmacists
Authors: Takahashi, et al.
We sincerely appreciate the positive, constructive, and encouraging comments regarding our study.
Point-by-point response to the reviewer's comments:
This is interesting brief report. Manuscript is well written. Title suggests that the role of pharmacists will be directly analyzed in presented study. Similarly, the aim of the study covers such suggestion. But the methods and results are not concentrated on that problem. This is study on risk factors control, so the results may give only a suggestion what may be improve in primary/secondary prevention. Is it retrospective registry study? This should be clarified in methods section.
Ans.) This is a cross-sectional study.
It will be interesting to see follow up data and how the CT result influence the further pharmacotherapy.
Ans.) We agree with the reviewer’s comment. Since this study is a cross-sectional, there were no information with regard to pharmacotherapy after CT results.
Tables and figures are informative and well prepared.
Ans.) Thank you for the reviewer’s comment.
Conclusions: no sure if phrase: “suggesting the need for pharmacists to proactively suggest the use of combination tablets to medical doctors” is optimal. This suggest need for whole system improvement including the pharmacists.
Ans.) We agree with the reviewer’s comment. We revised the conclusions: “The % CAD at the time of CCTA examination was high (49 %), and the three major lifestyle-related diseases were not well controlled. Furthermore, the percentages of patients who used a 2-drug combination in the HTN (+), DL (+) and DM (+) groups were 34 %, 2 % and 14 %, respectively. Thus, the percentage of patients who used a 2-drug combination was low, implying that pharmacists should proactively suggest the use of combination tablets to medical doctors. This study also suggests the need for improvements to the entire system by medical staff, including with regard to patient guidance and advice to medical doctors by pharmacists.” (Lines 341-348).
The references are appropriate.
Ans.) Thank you for the reviewer’s comment.

Reviewer 2 Report
Comments and Suggestions for Authors
It is gratifying to read that pharmacists are appreciated and used for their knowledge of medicines “The role of pharmacists is evolving and expected to expand”.
The number of patients (1,357) included is impressive and is well divided into groups. But, please mention the time interval for the CCTA examination at your university hospital.
The paragraph 54 -59 does not clearly say what the pharmacist does, please rewrite it so that it is understood exactly what are the pharmacists’ interventions.
Fig 2 - graphic HTN: the values for DBP are almost the same in group (-) vs group (+), please check the correct p value.
In the paragraph 61-63 “In this study, we aimed to clarify what kind of prescribing guidance pharmacists should give to medical doctors to prevent CAD before CCAT examination for patients undergoing CCTA to screen for CAD.” –where is this prescribing guidance mentioned?
You must mention exactly what was the intervention of the pharmacists in this study. Otherwise, the study presents the situation of patients with CAD potential, +/- with other associated pathologies and the situation regarding the pharmacological classes used. You must justify the title of the work by clearly specifying the pharmacists' activity.
Author Response
Response to Reviewer #2
Ref. No. pharmacy-2969262R1
Presence of lifestyle-related diseases at the time of coronary artery CT examination and trends in medication use: Role of pharmacists
Authors: Takahashi, et al.
We thank this reviewer very much for reviewing our manuscript. We appreciate the reviewer's kind suggestions and have revised our manuscript according to the reviewer's comments.
Point-by-point response to the reviewer's comments:
It is gratifying to read that pharmacists are appreciated and used for their knowledge of medicines “The role of pharmacists is evolving and expected to expand”.
The number of patients (1,357) included is impressive and is well divided into groups. But, please mention the time interval for the CCTA examination at your university hospital.
Ans.) CCTA examination is performed when a patient complains of some chest symptoms.
The paragraph 54 -59 does not clearly say what the pharmacist does, please rewrite it so that it is understood exactly what are the pharmacists’ interventions.
Ans.) We added the following sentences to Introduction: “Up until now, the main role of pharmacists was to prepare medicines according to medical doctors' instructions and to explain to patients how to take them properly and their side effects. However, the scope of a pharmacist’s work has expanded to include providing prescription advice to medical doctors.” (Lines 63-67).
Fig 2 - graphic HTN: the values for DBP are almost the same in group (-) vs group (+), please check the correct p value.
Ans.) We confirmed it. There was a significant difference in DBP.
In the paragraph 61-63 “In this study, we aimed to clarify what kind of prescribing guidance pharmacists should give to medical doctors to prevent CAD before CCAT examination for patients undergoing CCTA to screen for CAD.” –where is this prescribing guidance mentioned?
You must mention exactly what was the intervention of the pharmacists in this study. Otherwise, the study presents the situation of patients with CAD potential, +/- with other associated pathologies and the situation regarding the pharmacological classes used. You must justify the title of the work by clearly specifying the pharmacists' activity.
Ans.) We added the following sentences to study limitations: “In addition, this study did not involve pharmacists intervening in any way with medical doctors' prescriptions. Since this is a cross-sectional study, this report examined the possible roles of pharmacists with regard to their prescriptions. A large-scale prospective study will be needed to address these issues.” (Lines 334-338).

Reviewer 3 Report
Comments and Suggestions for Authors
The manuscript 'Presence of lifestyle-related diseases at the time of coronary artery CT examination and trends in medication use: Role of pharmacists' shows the importance of collaboration between physicians and pharmacists in prescribing medications for patients with risk factors for coronary artery disease, especially in patients with several different risk factors. The authors have illustrated how much additional work and collaboration with other professionals is still needed to prevent cardiovascular diseases in patients with various and multiple risk factors.
Based on the above, the authors should make minor corrections in their manuscript.
It is necessary to indicate the time period in which the research was conducted. Namely, the authors state in the Discussion that a pill containing a combination of two drugs (e.g., statin and ezetimibe, and DPP-41 and biguanide) was not available at the time of the study. However, the combination of these two drugs was taken by only 2% and 14% of patients, respectively.
In presenting the results in Figures 1 and 2, the authors should indicate, perhaps in the text of the results, how many patients there are with and without hypertension, how many with diabetes, and how many with dyslipidemia, in order to assess relationships, i.e., ratios between patient groups.
In the conclusion of the manuscript, the authors should state the percentage of patients with CAD and the percentage of patients using a 2-drug combination. They should also state the three main lifestyle-related diseases.
The full name must be written at the first mention of the abbreviation CKD (line 99).
Author Response
Response to Reviewer #3
Ref. No. pharmacy-2969262R1
Presence of lifestyle-related diseases at the time of coronary artery CT examination and trends in medication use: Role of pharmacists
Authors: Takahashi, et al.
We thank this reviewer very much for reviewing our manuscript.
Point-by-point response to the reviewer's comments:
The manuscript 'Presence of lifestyle-related diseases at the time of coronary artery CT examination and trends in medication use: Role of pharmacists' shows the importance of collaboration between physicians and pharmacists in prescribing medications for patients with risk factors for coronary artery disease, especially in patients with several different risk factors. The authors have illustrated how much additional work and collaboration with other professionals is still needed to prevent cardiovascular diseases in patients with various and multiple risk factors.
Based on the above, the authors should make minor corrections in their manuscript.
It is necessary to indicate the time period in which the research was conducted. Namely, the authors state in the Discussion that a pill containing a combination of two drugs (e.g., statin and ezetimibe, and DPP-41 and biguanide) was not available at the time of the study. However, the combination of these two drugs was taken by only 2% and 14% of patients, respectively.
Ans.) We added the following sentences to the Discussion: “Pharmacists should take these findings into consideration when giving advice. On the other hand, in Japan, single-pill combination tablets that combine two drugs (such as statin and ezetimibe, or DPP-41 and biguanide, etc.) were not available on the market at that time. While it was possible to administer two drugs in combination, if a single-pill combination tablet had been available, administration methods that combined two drugs may have become more popular.” (Lines 312-317).
In presenting the results in Figures 1 and 2, the authors should indicate, perhaps in the text of the results, how many patients there are with and without hypertension, how many with diabetes, and how many with dyslipidemia, in order to assess relationships, i.e., ratios between patient groups.
Ans.) We added the data in the text.
In the conclusion of the manuscript, the authors should state the percentage of patients with CAD and the percentage of patients using a 2-drug combination. They should also state the three main lifestyle-related diseases.
Ans.) We revised the following sentences in the Conclusions: “Furthermore, the percentages of patients who used a 2-drug combination in the HTN (+), DL (+) and DM (+) groups were 34 %, 2 % and 14 %, respectively. Thus, the percentage of patients who used a 2-drug combination was low, implying that pharmacists should proactively suggest the use of combination tablets to medical doctors. This study also suggests the need for improvements to the entire system by medical staff, including with regard to patient guidance and advice to medical doctors by pharmacists.” (Lines 342-348).
The full name must be written at the first mention of the abbreviation CKD (line 99).
Ans.) We corrected it.

Round 2
Reviewer 1 Report
Comments and Suggestions for Authors
I have no more comments
Author Response
Thank you for your reviewing.
Reviewer 2 Report
Comments and Suggestions for Authors
From reading the new manuscript, I notice that the authors did not sufficiently and clearly resolve the issues mentioned by me in the first report.
The paragraph 54 -59 does not clearly say what the pharmacist does, please rewrite it so that it is understood exactly what are the pharmacists’ interventions.
Your “Ans.) We added the following sentences to Introduction: “Up until now, the main role of pharmacists was to prepare medicines according to medical doctors' instructions and to explain to patients how to take them properly and their side effects. However, the scope of a pharmacist’s work has expanded to include providing prescription advice to medical doctors.” (Lines 63-67).”
If the study only shows what clinical pharmacists could do, then the title is misleading, because we expected to read about the pharmacist's role in practice. The present study indicates that a clinical pharmacist would be needed, but the clinical pharmacist was not actively involved in the presented results.
Fig 2 - graphic HTN: the values for DBP are almost the same in group (-) vs group (+), please check the correct p value.
Your “Ans.) We confirmed it. There was a significant difference in DBP.”
The confirmation must be accompanied by an explanation, please argue with experimental data, mean, SD
You must mention exactly what was the intervention of the pharmacists in this study. Otherwise, the study presents the situation of patients with CAD potential, +/- with other associated pathologies and the situation regarding the pharmacological classes used. You must justify the title of the work by clearly specifying the pharmacists' activity.
Your Ans.) “We added the following sentences to study limitations: “In addition, this study did not involve pharmacists intervening in any way with medical doctors' prescriptions. Since this is a cross-sectional study, this report examined the possible roles of pharmacists with regard to their prescriptions. A large-scale prospective study will be needed to address these issues.” (Lines 334-338).”
The “Role of pharmacists” from the title “Presence of lifestyle-related diseases at the time of coronary artery CT examination and trends in medication use: Role of pharmacists” is not justified by the presented research.
Author Response
Response to Reviewer #2
Ref. No. pharmacy-2969262R2
Considering the possible role of pharmacists according to the presence or absence of lifestyle-related diseases at the time of coronary CT examination and trends of medication use for these diseases by medical doctors
Authors: Takahashi, et al.
We thank this reviewer very much for reviewing our manuscript.
Point-by-point response to the reviewer's comments:
The paragraph 54 -59 does not clearly say what the pharmacist does, please rewrite it so that it is understood exactly what are the pharmacists’ interventions. Your “Ans.) We added the following sentences to Introduction: “Up until now, the main role of pharmacists was to prepare medicines according to medical doctors' instructions and to explain to patients how to take them properly and their side effects. However, the scope of a pharmacist’s work has expanded to include providing prescription advice to medical doctors.” (Lines 63-67).” If the study only shows what clinical pharmacists could do, then the title is misleading, because we expected to read about the pharmacist's role in practice. The present study indicates that a clinical pharmacist would be needed, but the clinical pharmacist was not actively involved in the presented results.
Ans.) We agree with you. We changed the title to “Considering the possible role of pharmacists according to the presence or absence of lifestyle-related diseases at the time of coronary CT examination and trends of medication use for these diseases by medical doctors” (Lines 2-4) and background to “Because patients often already have coronary artery disease (CAD) at the time of a coronary artery computed tomography angiography (CCTA) examination, we examined the medications prescribed by medical doctors for lifestyle-related diseases and investigated what possible role of pharmacists can play in prescribing.” (Lines 17-20 and 90-92).
Fig 2 - graphic HTN: the values for DBP are almost the same in group (-) vs group (+), please check the correct p value. Your “Ans.) We confirmed it. There was a significant difference in DBP.” The confirmation must be accompanied by an explanation, please argue with experimental data, mean, SD
Ans.) We have already shown the DBP (mean±SD) in second version and written that there was a significant difference (Lines 211-213).
You must mention exactly what was the intervention of the pharmacists in this study. Otherwise, the study presents the situation of patients with CAD potential, +/- with other associated pathologies and the situation regarding the pharmacological classes used. You must justify the title of the work by clearly specifying the pharmacists' activity. Your Ans.) “We added the following sentences to study limitations: “In addition, this study did not involve pharmacists intervening in any way with medical doctors' prescriptions. Since this is a cross-sectional study, this report examined the possible roles of pharmacists with regard to their prescriptions. A large-scale prospective study will be needed to address these issues.” (Lines 334-338).” The “Role of pharmacists” from the title “Presence of lifestyle-related diseases at the time of coronary artery CT examination and trends in medication use: Role of pharmacists” is not justified by the presented research.
Ans.) We agree with you the title was not justified by the presented research and changed the title and background (Lines 2-4, 17-20 and 90-92).
Round 3
Reviewer 2 Report
Comments and Suggestions for Authors
I hope that in the future you will publish articles about the activity of the clinical pharmacists.